# An Analysis of the Correct Frequency of the Service Inspections of German Passenger Cars—A Case Study on Kazakhstan and Poland

Saltanat Nurkusheva [1,2], Michał Bembenek [3,*], Maciej Berdychowski [4], Bożena Gajdzik [5,*] and Radosław Wolniak [6]

1 Department of Organization of Transport, Traffic and Transport Operations, L.N. Gumilyov Eurasian National University, Satbaev 2, Astana 010000, Kazakhstan; saltanat.nurkusheva@kazatu.kz
2 Department of Transport Equipment and Technologies, S. Seifullin Kazakh Agrotechnical Research University, Zhenis 62B, Astana 010011, Kazakhstan
3 Faculty of Mechanical Engineering and Robotics, AGH University of Krakow, A. Mickiewicza 30, 30-059 Krakow, Poland
4 Faculty of Mechanical Engineering, Poznan University of Technology, 5 M. Skłodowska-Curie Square, 60-965 Poznan, Poland; maciej.berdychowski@put.poznan.pl
5 Department of Industrial Informatics, Silesian University of Technology, 44-100 Gliwice, Poland
6 Faculty of Organization and Management, Silesian University of Technology, 44-100 Gliwice, Poland; radoslaw.wolniak@polsl.pl
* Correspondence: bembenek@agh.edu.pl (M.B.); bozena.gajdzik@polsl.pl (B.G.)

**Abstract:** This article presents a case study on estimating the real service inspection intervals for German-brand passenger cars in Kazakhstan and Poland. This study aimed to identify disparities between the official recommendations of manufacturers for car maintenance and the real data collected in these two countries. The following passenger cars were examined: Audi A6, Q5, and Q8; Porsche Cayenne and Cayenne coupe; and Volkswagen Passat, Polo, Teramont, Tiguan, Touareg, Arteon, Golf, T-Cross, Tiguan all space, Touran, T-Roc, and Up. To assess the difference between real and recommended values, the manufacturer criteria of a recommended mileage of 15,000 and 30,000 km or a time frame of 365 and 730 days to the first service inspection were applied. The data analysis showed that in Kazakhstan, 31.4% of cars did not meet the warranty conditions, while in Poland, it was 21.0%. The dominant criterion that was not met was the time criterion. The assessment of these factors emphasizes the importance of customizing vehicle maintenance schedules to the specific conditions and driving behaviors prevalent in each country. The practical contribution of the article lies in uncovering the discrepancies between official manufacturer recommendations for car maintenance and the actual data collected in Kazakhstan and Poland. By identifying specific models, Volkswagen Touareg and Tiguan in Kazakhstan and Volkswagen Up in Poland, for which the maintenance intervals deviated significantly from those recommended, this study offers valuable insights for optimizing service schedules and improving the efficiency of maintenance practices in these countries. From a scientific perspective, this article contributes by providing empirical evidence of real-world maintenance behaviors for German-brand passenger cars.

**Keywords:** car service inspection; frequency of car service inspection; analysis of car service inspection; improvement of time between inspections





## 1. Introduction

The maintenance of a new passenger car is a process that includes the regular technical inspection of the vehicle to detect and eliminate faults [1]. It is also preventive maintenance to maintain the vehicle's performance and keep all the installed systems in operation [2]. Routine maintenance usually includes regular inspection and the service of the equipment, which is performed according to the manufacturer's recommendations. Properly carried

out maintenance schedules help in the early detection and resolution of potential issues [3,4], can prevent serious breakdowns [5], and keep warranty obligations [6,7]. It is also worth noting that conducting regular technical service inspections according to the manufacturer's recommendations is a good practice to ensure the proper functionality and safety of the vehicle [8] and, above all, the safety of the passengers travelling in it. Hence, it is essential to establish a suitable maintenance schedule for the vehicles, to maintain a working, robust, and reliable transportation system [9]. The adjustment of the frequency of service inspection (SI) is one of the methods for managing the technical conditions of cars and equipment and is the process of determining the optimal SI intervals, considering the operational characteristics of the vehicles [10] and conditions in which the vehicle is operating. Establishing an effective inspection schedule is a critical aspect. It is worth emphasizing that too frequent and poorly selected service intervals unnecessarily expose car users to additional costs of time and money [11]. Overly long service intervals increase the likelihood of failure to detect faults and can ultimately lead to serious mechanical failures [1].

Traditional approaches to determining the optimal inspection interval often rely on a fixed reassessment interval or mileage that remains constant throughout the entire service period. However, there is a need for alternative methods that offer more flexibility and adaptability in optimizing inspection intervals. The main goals of adjusting the frequency of SI are to increase the reliability and safety of equipment operation, to reduce the cost of its operation and maintenance, and to increase the interval between major repairs. The determination of the optimal SI intervals is based on the analysis of data obtained during the operation of the vehicle and its equipment [12], as well as on the use of methods for predicting the technical condition [13]. To determine the optimal preventive maintenance checkpoints, a simulation analysis [14,15], evolutionary algorithm [16], or genetic algorithm [17] can be used, e.g., in work [18], an algorithm for automating multidimensional data analysis was presented. The issues of car maintenance have also been addressed in [19], where dynamic programming was utilized to reduce downtime, while software [20,21] has been employed to streamline operations. This maintenance strategy ensures a consistent level of quality and safety within the transportation system. SIs are not always carried out correctly. For example, in [22], an analysis of the reliability of car safety was carried out. In such cases, more frequent technical servicing may be recommended to maintain the reliability and safety of the vehicle. It is also worth emphasizing that the maintenance strategy ensures a consistent level of quality and safety not just for an individual car, but also within the transportation system if we are dealing with fleet cars [23].

The operating conditions of passenger cars may vary. First, vehicles can be operated in different climates, which is reflected, for example, in the selection of the type of engine oil [24]. The ambient temperature of car operation primarily affects the wear of the engine (cold start [25]), transmission system (e.g., high oil viscosity in gears [26]), and suspension components (especially parts containing rubber and oils, e.g., shock absorbers [27]). Precipitation and high humidity cause accelerated corrosion of the body and chassis components [28]. Another element that may influence the service interval is the condition of the roads—not only in terms of the condition of their surfaces, but also in how they are maintained in winter [29] and the distance traveled between each engine start [30]. The choice of Poland and Kazakhstan as the focuses of this study is strategically justified based on their geographical locations and representation of distinct regions. Kazakhstan, situated in the heart of Asia, offers a unique perspective on maintenance practices and intervals within a Central Asian context. On the other hand, Poland, located in the heart of Europe, serves as a representative of Central European countries. In terms of new vehicles sold in Kazakhstan, for the Volkswagen Audi Gruppe (VAG) group, it is recommended to conduct a technical inspection of a passenger car every 15,000 km or once a year [31]. However, in Poland, a different approach is adopted for these types of vehicles, and it is recommended to carry out a technical inspection of the car every 30,000 km or once every two years. These recommendations are based on various factors such as operating

conditions [32], road quality [33], and technical requirements [34]. In Kazakhstan [35,36], the vehicle is operated under conditions that can affect its performance and wear: an extreme, continental climate [37], relatively poor road conditions, low-quality fuel [38], and circumstances promoting corrosion [39]. It may be advisable to perform maintenance before the mileage specified by the manufacturer [40]. Additionally, car owners make heavy use of their vehicles, by transporting cargo, towing heavy loads [41], and undertaking constant long-distance trips, which increases the strain on the vehicle's components. In comparison, Poland also experiences varying conditions that affect vehicle operation. For instance, Poland has a temperate climate with four distinct seasons. Road conditions in Poland can vary, ranging from well-maintained highways to rural roads that may be less smooth. The quality of fuel in Poland generally meets European standards. In Poland, similarly to Kazakhstan, the maintenance of passenger cars involves regular technical inspections, diagnostics [42,43], and preventive maintenance to ensure safety and performance [44].

The digitization of industry [45], including maintenance processes, has been a significant focus in recent years [46–48]. By leveraging data analytics, Internet of Things devices, and automation, organizations can achieve higher levels of efficiency and productivity in maintenance operations at work [49]. In the context of road transport companies, Caban et al. conducted statistical analyses of maintenance parameters for vehicles [32]. Their research emphasizes the importance of analyzing maintenance data to identify trends, optimize SI schedules, and reduce operational costs [50]. By leveraging statistical models, organizations can make data-driven decisions and allocate resources more efficiently. Despite access to highly sophisticated methods of analysis of operational data, statistical data are still the basic analytic tool [51,52], upon which extensive control systems for operation and maintenance can be built.

The article analyzes the dates of the first SI of new passenger cars from the VAG group. The research aimed to examine the correctness of service recommendations regarding the date of the first inspection and to demonstrate differences regarding the date of the first inspection for vehicles operated in completely different climatic and road conditions. By including data from both Kazakhstan and Poland, a broader perspective on SI frequencies and patterns was obtained. This comparative analysis allows for a better understanding of potential differences or similarities in maintenance practices between the two countries. It ultimately is shown that the time of the first inspection should also depend on the brand and model of the car.

The scientific contribution of this article lies in its investigation of the discrepancies between official manufacturer recommendations for car maintenance and the actual data collected in Kazakhstan and Poland. By analyzing real-world maintenance behaviors for specific German-brand passenger cars in these two countries, the study offers valuable insights into the practical application of maintenance schedules and the factors influencing adherence to manufacturer guidelines. Furthermore, by focusing on a range of popular models and considering regional differences in driving conditions and maintenance practices, the research contributes to our understanding of how geographical and cultural factors can impact maintenance intervals.

## 2. Materials and Methods

### 2.1. Materials

In total, data on 133 vehicles were analyzed. The research focused solely on passenger cars intended for personal use in Kazakhstan and Poland to investigate the real SI intervals compared to manufacturers' recommendations, revealing notable discrepancies in maintenance practices. The Elsa Win program was used to collect the data. The data analyzed come from the following two databases: Volkswagen Center Astana dealership database (57 cars) and Wielkopolska Volkswagen Center dealership database (76 cars). In the case of data from Astana, Kazakhstan, the following car brands and models were analyzed: Audi A6, Audi Q5, Audi Q8, Porsche Cayenne, Porsche Cayenne Coupe, Volkswagen Passat, Volkswagen Polo, Volkswagen Teramont, Volkswagen Tiguan, and Volkswagen Touareg

(Table A1, Appendix A). In the case of data from Wielkopolska, Poland, the following car brands and models were analyzed: Volkswagen Arteon, Volkswagen Golf, Volkswagen Passat, Volkswagen T-Cross, Volkswagen Tiguan, Volkswagen Tiguan allspace, Volkswagen Touran, Volkswagen T-Roc, and Volkswagen Up (Table A2, Appendix A). The collected data include the following information: car brand, car model, date of sale, mileage to the first SI, and date of the first SI.

### 2.2. The Data Analysis

The first step of data analysis was to determine the number of days that had passed from the purchase of the car to its first technical inspection. In terms of new vehicle sales in Kazakhstan, for the VAG group, it is recommended to conduct a technical inspection of a passenger car every 15,000 km or once a year. However, in Poland, a different approach is adopted for VAG vehicles, with a recommendation to carry out a technical inspection of the car every 30,000 km, or once every two years. An initial difficulty in comparing the data was the different recommended intervals and service intervals. Hence, to better interpret the data, the information obtained on the biennial service interval for Polish cars was also presented on an annual basis. Mileage data in 5000 km intervals and the first inspection in semi-annual intervals are presented. Data from individual models were also analyzed. The arithmetic mean of the mileage and the time elapsed until the first inspection was determined for each model.

### 2.3. Fulfillment of Warranty Conditions

Determining which car owners met the recommended inspection intervals was carried out by analyzing the available data at the time of the first inspection and the mileage reached during this period. The inspection was deemed to have taken place on time if the car mileage was not more than 15,000 km in Kazakhstan and the first inspection took place within 365 days of the date of purchase, and in Poland, if the vehicle mileage was not more than 30,000 km and the first inspection took place within 730 days of the date of purchase. If the warranty conditions were not met, the criteria that were not fulfilled were examined. This included checking whether the recommended period for the first inspection, the mileage of the vehicle, or both criteria had not been met.

### 2.4. The Algorithm for Calculating the Proper Date of Service Inspection

One of the goals of this article was also to analyze whether car owners carried out the first SI within the period and mileage specified by the manufacturer, or whether SI should be carried out earlier or later. To analyze the adjustment of the frequency of SI of the cars, the following algorithm was used:

1. Calculation of average mileage for each brand and type of vehicle.
2. Calculation of average days to first SI.
3. Calculation of the hypothetical number of days when the car will reach the mileage recommended by the manufacturer for the first inspection (15,000 km for Kazakh cars or 30,000 km for Polish cars).
4. Determining the criteria of *time* or *mileage*—which involves determining which of the parameters will be exceeded first—days (365 days in the case of Kazakh cars and 730 days for Polish cars), or the hypothetical exceeding of the mileage limit (15,000 km for Kazakh cars or 30,000 km for Polish cars).
5. Determining the correct inspection time (the difference in days between when the inspection should be carried out as opposed to when the inspection was actually performed) according to the proper criterion. If the time criterion was selected, the average number of days had to be subtracted from the prescribed service deadlines: 365 days for Kazakh vehicles and 730 days for Polish vehicles. If the mileage criterion was selected, a proportional calculation of the number of days to reach the recommended mileage (15,000 km in the case of Kazakh cars, 30,000 km in the case of Polish cars) was performed.

*2.5. Determining Weather Conditions in Kazakhstan and Poland*

Weather conditions were determined using data from the Weather Spark website [53]. Average daily maximum and minimum temperatures for the years 2021, 2022, and 2023 were collected and used to determine the mean maximum and minimum temperatures over the three years considered. Moreover, the corresponding averages and extremes of the maximum and minimum temperatures were identified. The charts show average minimum, maximum, average monthly ambient temperatures, and precipitation in Astana, Kazakhstan and Poznań, Poland. The obtained climatic data were also averaged over the entire calendar year. The data for each month were averaged from the years 2021 to 2023.

## 3. Results and Discussion

*3.1. Analysis of Mileage Data and Time of the First Inspection in Kazakhstan and Poland*

Figure 1a,c show each car's mileage at the time of the first inspection in its respective country (according to Tables A1 and A2 in Appendix A). To make the data easier to interpret, the data for cars from Poland are presented in a format measuring half the time between inspections, i.e., one year. The average mileage until the first inspection for cars in Kazakhstan was 9526 km, while the average annual mileage for Polish cars (concerning the half-cycle between inspections) was 10,029 km. This is interesting, as in Kazakhstan, the distances between cities are much greater than in Poland. However, this can be explained by the condition of the road surfaces. According to the data by Shakhmov and Hafiz (2021) [36] in Kazakhstan, the road quality is at times worse than in Poland. Additionally, European transportation infrastructure is sufficiently developed, as evidenced by Hussain et al. (2020) [53], which may favor longer car journeys. Figure 2 shows the distribution of car mileage in intervals of 5000 km.

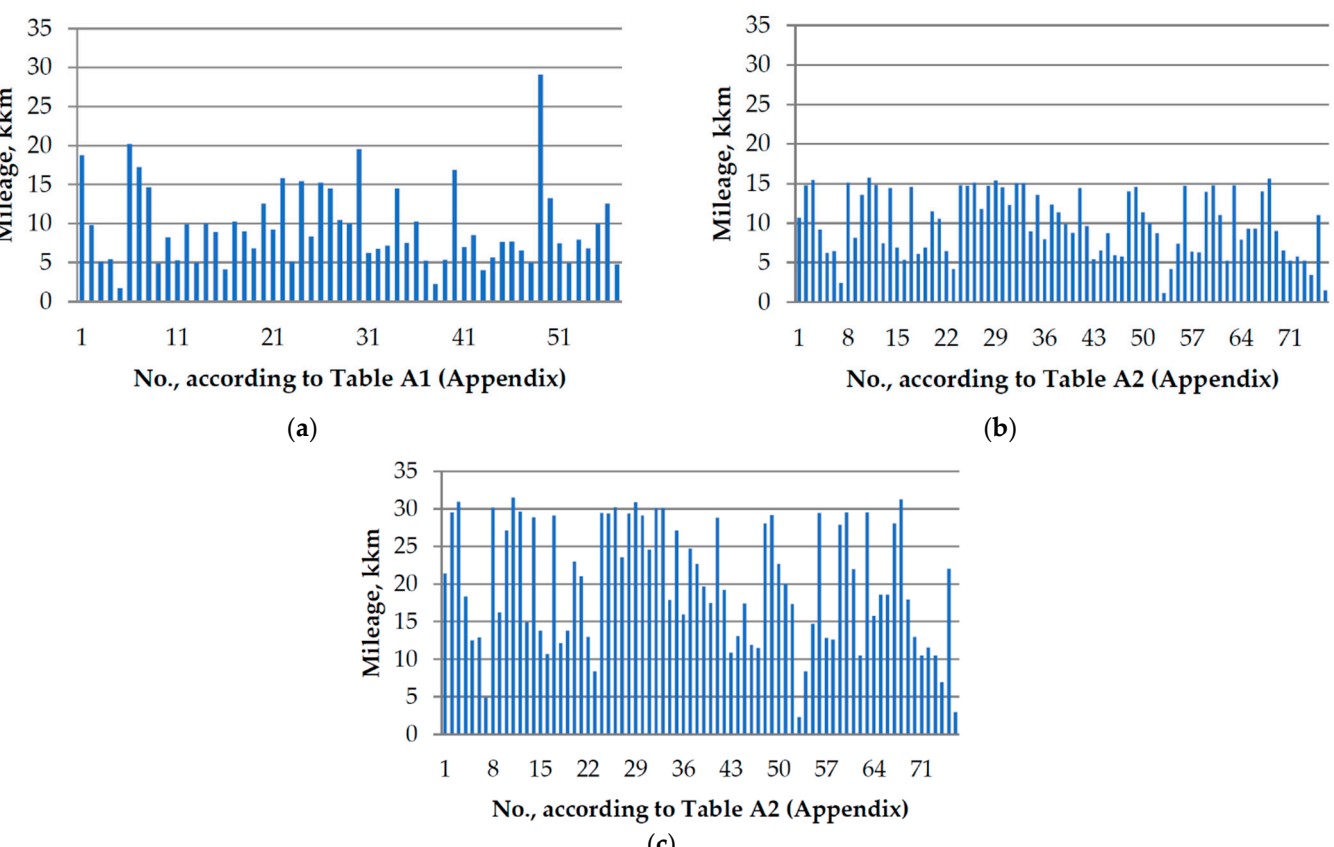

**Figure 1.** The bar chart of the mileage at SI-1 (according to Tables A1 and A2 in Appendix A): (**a**) Kazakhstan; (**b**) Poland considering the half-cycle between inspections, PL/2—Poland; (**c**) Poland.

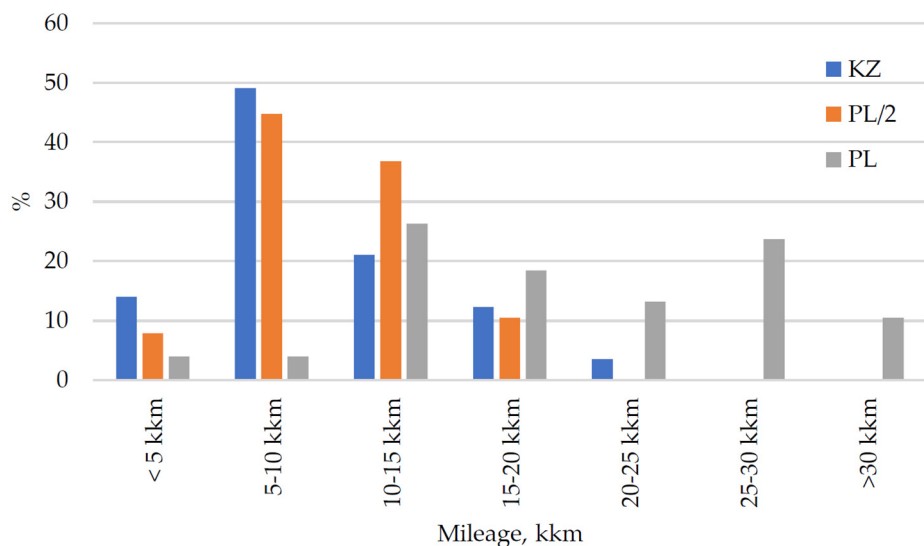

**Figure 2.** Mileage percentage intervals of 5000 km: KZ—Kazakhstan; PL/2—Poland considering the half-cycle between inspections; PL—Poland.

The chart visually illustrates the difference in mileage between Kazakhstan and Poland. From the chart, it can be seen that in Kazakhstan, there are twice as many cars with a mileage not exceeding 5000 km per year. In the range between 5000 and 10,000 km, the results are similar, with a slight advantage for Kazakh cars. The situation is completely different in the next range of 10,000 to 15,000, where Polish cars dominate. The number of compartments covering the distance between 15,000 and 20,000 for cars from both countries is similar.

Figure 3 shows the percentage of cars that exceeded the criteria for the first inspection recommended by the manufacturer. In Kazakhstan, 31.4% of cars did not meet the warranty conditions, while in Poland it was 21.0%. The dominant criterion that was not met was the time criterion, amounting to between 10.5% and 15.6% of the total number of cars at the time of the first technical inspection.

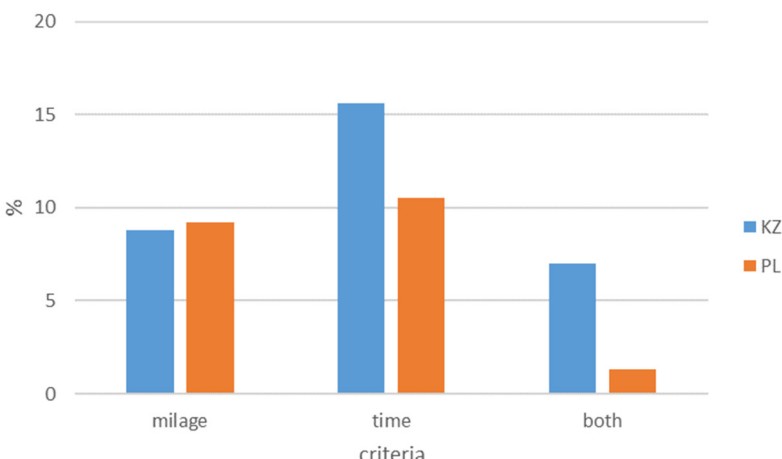

**Figure 3.** Percentage of cars that did not meet the warranty criteria: KZ—Kazakhstan; PL—Poland.

Figure 4 presents a histogram illustrating the distribution of days between the purchase date and the SI of the car. The histogram is derived from the analysis conducted on the data collected from Tables A1 and A2 (Appendix A). Figure 5 displays a histogram that explores five different periods when the first technical inspection was conducted: within six months, one year, and beyond.

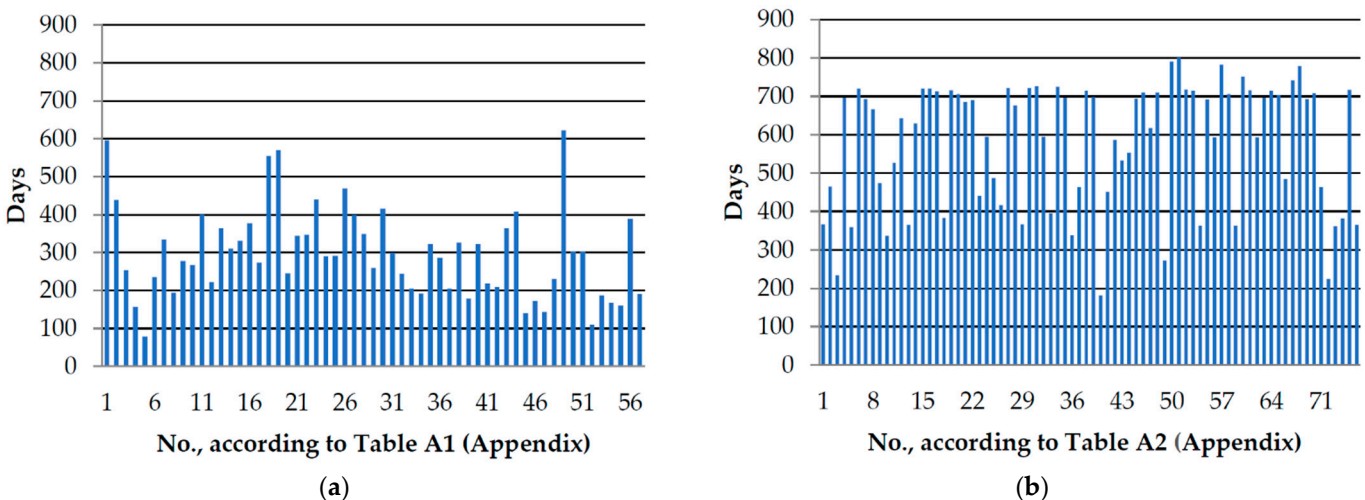

**Figure 4.** The bar chart of the days between the purchase date and the SI-1 (according to Tables A1 and A2 in Appendix A): (**a**) Kazakhstan; (**b**) Poland.

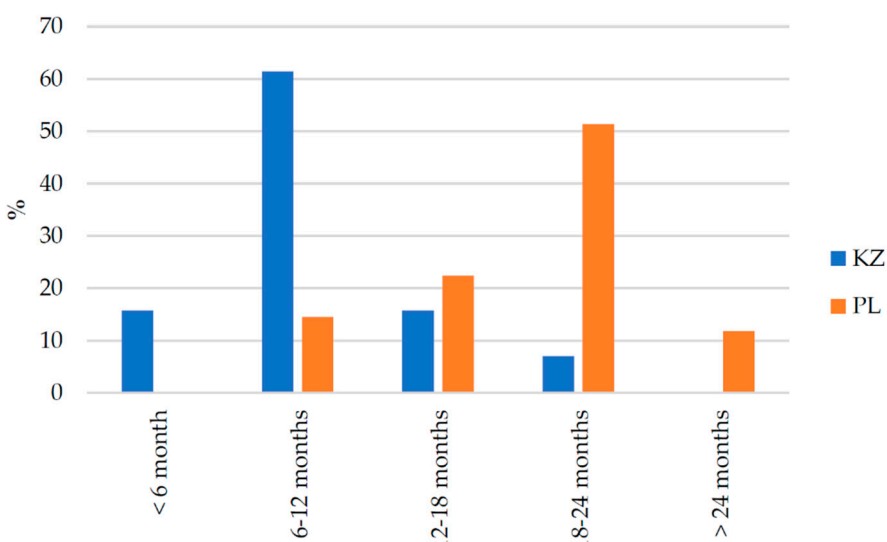

**Figure 5.** The quantity of SI in intervals of 6 months: KZ—Kazakhstan; PL—Poland.

The histogram in Figure 5 clearly illustrates the difference in the timing of the first SI for cars from Kazakhstan and Poland. In numerical terms, within the period of up to 6 months, there are nine vehicles, from 6 to 12 months there are thirty-five cars, from 12 to 18 months there are nine cars, and from 18 to 24 months there are four cars from Kazakhstan. In contrast, for cars from Poland, there are eleven vehicles from 6 to 12 months, seventeen vehicles from 12 to 18 months, thirty-nine vehicles from 18 to 24 months, and nine vehicles that have undergone the first maintenance after more than 24 months.

The average mileage for the first technical service (Figure 6a) falls within the range of 6462 to 18,738 km depending on the car type from Kazakhstan. The minimum mileage is observed in Volkswagen Teramont, at 57% lower than the value of 15,000 km. Polo and Tiguan have lower mileage by 32–34%, Passat by 25%, Touareg by 39%, and Audi A6 and Q8 by 45%. Only Audi Q5 shows an increase of 25% in mileage. In Poland, the average mileage on the first technical service (Figure 6c) ranges from 10,616 to 25,926 km, depending on the car model. The lowest mileage is observed for the Volkswagen Up, at 29% lower than the value of 15,000 km. The highest mileage is recorded for the Tiguan at 25,926 km and the Arteon at 25,464 km.

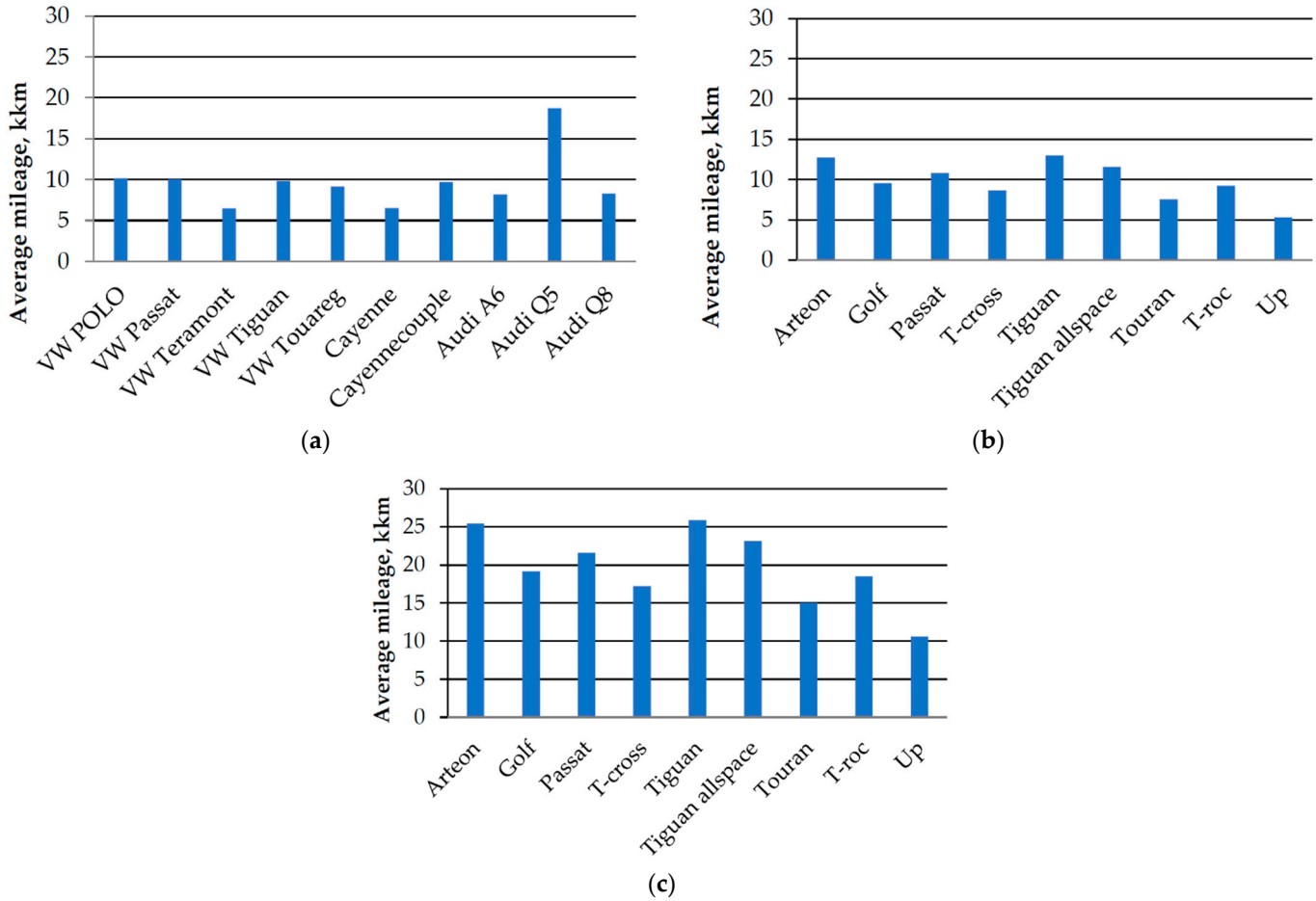

**Figure 6.** Bar chart of average mileage value depending on the car type (according to Tables A1 and A2 in Appendix A): (**a**) Kazakhstan; (**b**) Poland considering the half-cycle between inspections, PL/2—Poland; (**c**) Poland.

When comparing the average mileage values for car models operated in Kazakhstan and Poland, it is evident that there are notable differences. According to the data from Figure 6, the annual mileage values in Kazakhstan and half of the biennial values in Poland (Pl/2) reveal the minimum average mileage in Poland to be 5308, while in Kazakhstan, it stands at 6462.75. On the other hand, the maximum average mileage values in Poland amount to 12,963, whereas in Kazakhstan, the figure reaches 18,738.5. Accounting for all the values of car models and calculating the overall average, the average mileage in Kazakhstan is 9703.2, while in Poland it is 9816.1.

*3.2. Analysis of the Proper Date of SI in Kazakhstan and Poland*

Upon analyzing the vehicles using the provided data from Table A1 in Appendix A, the following observations can be made. After purchasing a car, the first technical maintenance is performed after a certain number of days. It is worth noting that the average mileage of the car models in the database is below 15,000 km. If we consider the recommended service interval of 15,000 km, it is possible to calculate the interval in days for the first maintenance based on the calculation model. However, based on the average values, the difference in days between the average and estimated intervals ranges from −57 to 118 days. Negative values, such as −57, suggest shortening the maintenance interval by 57 days, while positive values, such as 118, indicate a possible extension of the maintenance interval by 118 days. All the obtained values are presented in Table 1. As for the Polish cars, the interval fluctuates from 74 to 265 days, based on the data extracted from Table A2 in Appendix A. All the obtained values are presented in Table 2.

**Table 1.** Average mileage and interval of Kazakhstan cars at first technical service.

| Brand | Model | Average Mileage to First SI, km | Average Days to First SI | Estimated Days to Reach 15,000 km | Method of Criteria, Mileage (15,000 km) or Days (365 Days) | The Difference in Days between the Average and Estimated |
|---|---|---|---|---|---|---|
| Audi | A6 | 8163 | 304 | 559 | days | 61 |
| | Q5 | 18,739 | 285 | 228 | mileage | −57 |
| | Q8 | 8286 | 291 | 527 | days | 74 |
| Porsche | Cayenne | 6518 | 354 | 815 | days | 11 |
| | Cayenne coupe | 9666 | 411 | 638 | days | −46 |
| Volkswagen | Polo | 10,157 | 293 | 433 | days | 72 |
| | Passat | 10,062 | 377 | 562 | days | −12 |
| | Teramont | 6463 | 280 | 650 | days | 85 |
| | Tiguan | 9863 | 248 | 377 | days | 117 |
| | Touareg | 9116 | 247 | 406 | days | 118 |

**Table 2.** Average mileage and interval of Polish cars at first technical service.

| Brand | Model | Average Mileage to First SI, km | Average Days to First SI | Estimated Days to Reach 30,000 km | Method of Criteria, Mileage (30,000 km) or Days (730 Days) | The Difference in Days between the Average and Estimated |
|---|---|---|---|---|---|---|
| Volkswagen | Arteon | 25,464 | 416 | 490 | mileage | 74 |
| | Golf | 19,149 | 522 | 818 | days | 208 |
| | Passat | 21,586 | 601 | 835 | days | 129 |
| | T-Cross | 17,223 | 620 | 1080 | days | 110 |
| | Tiguan | 25,926 | 584 | 676 | mileage | 92 |
| | Tiguan allspace | 23,177 | 317 | 410 | mileage | 93 |
| | Touran | 15,049 | 560 | 1116 | days | 170 |
| | T-Roc | 18,499 | 633 | 1026 | days | 97 |
| | Up | 10,616 | 465 | 1314 | days | 265 |

### 3.3. Analysis of the Impact of Weather Conditions and Other Factors on the Frequency of First Inspections in Kazakhstan and Poland

The climate in a particular region plays a significant role in the wear and tear experienced by vehicles [54]. Extreme temperatures, whether hot or cold, can affect the performance and longevity of various vehicle components. The climate in Astana, Kazakhstan exhibits a significant temperature range in Figure 7. When comparing the climate conditions in Astana and Poznań, noticeable differences become apparent. In January, the average maximum temperature over three years in Astana is −10.7 °C, with a minimum of −20 °C, while Poznań experiences an average maximum temperature of 1.7 °C and a minimum of −3.3 °C. Moving to February, Astana records an average maximum temperature of −10 °C and a minimum of −19 °C, while Poznań sees an average maximum temperature of 3.3 °C and a minimum of −3.3 °C.

This indicates that winter temperatures in Poznań are significantly higher than those in Astana.

Examining the spring months, in May, the average maximum temperature in Astana is 20 °C, with a minimum of 7.7 °C, while Poznań experiences average maximum temperatures of 18.7 °C and minimum temperatures of 8 °C. This trend continues into April and March, where Poznań generally maintains milder temperatures compared to Astana.

As for the summer months, in June and July, Astana experiences average maximum temperatures of 25 °C and 26.7 °C, respectively, with corresponding minimum temperatures of 12.3 °C and 12.7 °C. In contrast, Poznań records average maximum temperatures of 22 °C

and 24 °C, with minimum temperatures of 11 °C and 13.7 °C. The temperature difference is less pronounced during the summer.

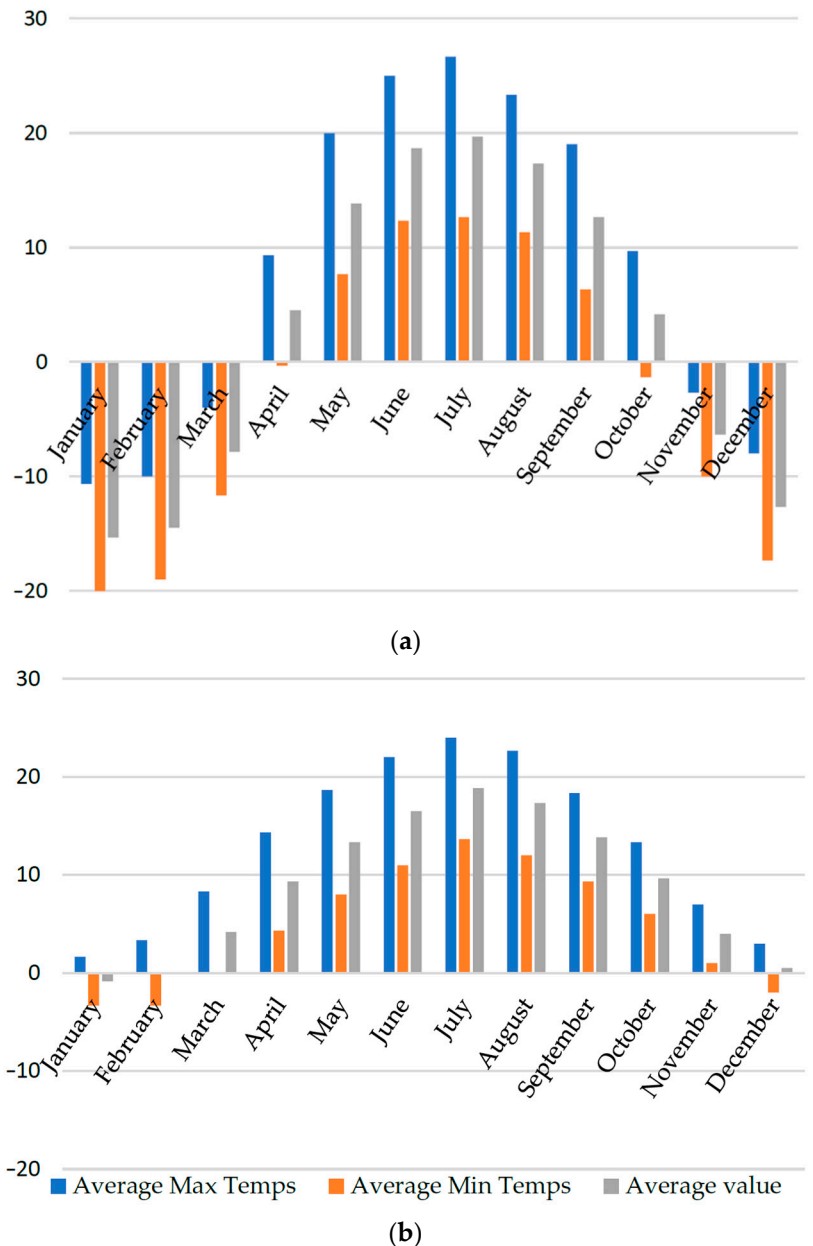

**Figure 7.** Bar chart of average temperature: (**a**) in Kazakhstan; (**b**) in Poland.

Considering the winter and autumn periods, which encompass December, January, and February, and September, October, and November, respectively, Poznań consistently exhibits milder temperatures compared to Astana.

Furthermore, it is notable that the results for the maintenance interval values between the average and estimated values in Kazakhstan encompass both negative and positive numbers, while in Poland, only positive figures are observed.

As noted earlier, climate directly influences the wear and tear experienced by vehicles, with extreme temperatures accelerating the degradation of parts. In regions like Astana, Kazakhstan, known for harsh winters, vehicles may require more frequent maintenance due to the effects of extreme cold on components like batteries, fluids, and seals. The Volkswagen Touareg and Tiguan exhibited significant deviations from optimal values, suggesting the potential for extended service intervals and earlier technical maintenance.

On the other hand, milder climates like that in Poznań, Poland, may result in less severe wear and tear, allowing for longer maintenance intervals. In our findings, the Volkswagen Up model could be extended by 265 days and the Volkswagen Arteon by 74 days due to the more moderate climate conditions. When comparing the findings with those of [55], similar conclusions were reached regarding the detrimental effects of unfavorable weather conditions on the performance of road vehicles. Conducting weather tests on vehicles is vital for assessing how precipitation influences driver visibility, sensor signals, tire traction, and structural integrity in the face of corrosion, all of which are essential for ensuring safety.

Humidity levels in each area can have a notable impact on vehicle maintenance as well. Higher humidity levels can contribute to increased corrosion of metal components, degradation of electrical connections, and potential damage to the vehicle's exterior and interior surfaces. Humidity likewise impacts the stability of electric sensors, as with the increase in ambient humidity, the response of the sensor attenuates rapidly and the phase shift increases rapidly [56]. In regions with elevated humidity, greater attention to rust prevention, interior conditioning, and electrical system maintenance may be necessary to ensure the long-term health of the vehicle. Furthermore, the humidity, as indicated by the number of days with precipitation, is higher in Poznań than in Astana in Figure 8. For instance, in July, days of precipitation were measured at 8.3 in Poznań and 6.4 in Astana. The higher frequency of rainy days in Poznań indicates a more humid climate throughout the year. This suggests a greater need in Poznań for specialized maintenance practices to mitigate the effects of moisture on vehicle components. This is because greater attention to rust prevention and electrical system maintenance may be necessary to protect the vehicle from the negative effects of humidity. When comparing the results with those in [57], it is evident that they are similar to the findings in the study that humidity is an important factor influencing the stability of the electronic system. The study in [58] found that as air humidity goes up from 0% to 60%, the critical electrical field where breakdown occurs increases by 1.22 kV/cm.

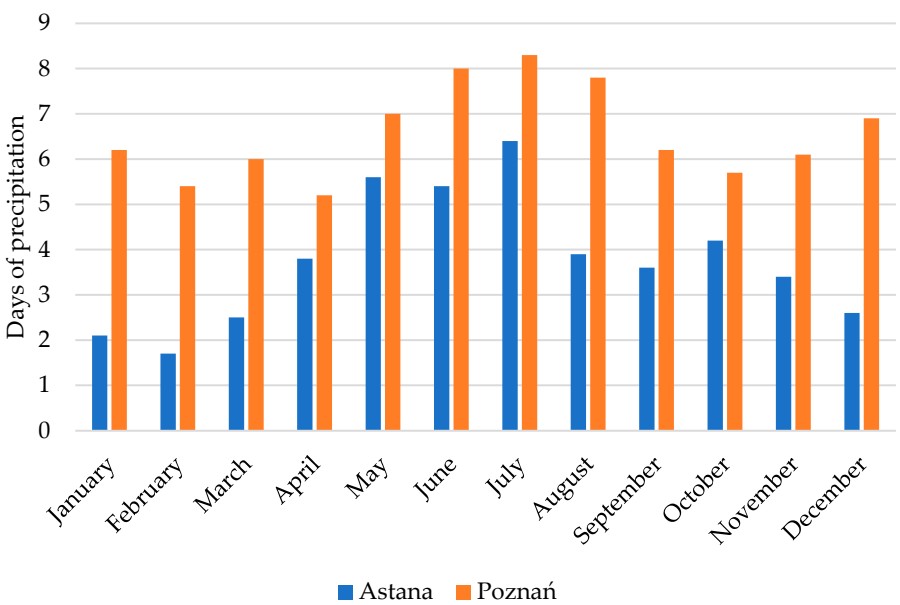

**Figure 8.** Bar chart of days with precipitation.

Driving style encompasses factors such as acceleration habits, braking tendencies, adherence to speed limits, and overall vehicle handling. Aggressive driving habits, frequent abrupt acceleration and braking, and high-speed driving can contribute to increased wear and tear on the vehicle, potentially necessitating more frequent SIs and maintenance. Conversely, smooth and consistent driving practices can contribute to prolonged intervals between SIs and maintenance requirements. The results of the study in [59] indicated that

drivers can influence the condition of their vehicles by adjusting their driving style, even with identical mileage and maintenance practices.

Polish drivers are generally considered to be more restrained and predictable behind the wheel. They tend to adhere to traffic regulations and are less likely to be involved in traffic accidents [60]. Drivers in Poland also tend to be more attentive to pedestrians and other road users.

In contrast, drivers in Kazakhstan may exhibit a more aggressive driving style, characterized by dynamic driving [61], frequent maneuvers, and less strict adherence to traffic rules [62]. The analysis findings revealed that in Kazakhstan, 31.4% of vehicles did not meet the warranty requirements, compared to 21.0% in Poland.

It is important to note that these are general tendencies, and each driver has a unique driving style. Considering the relationship between driving style and vehicle maintenance, it is apparent that an individual's manner of driving can significantly impact the wear and tear experienced by a vehicle, subsequently influencing maintenance requirements. Moreover, the minimal variances in the real SI time and estimated SI are clearly exemplified by the Porsche Cayenne Coupe and Audi Q5. Specifically, the analysis suggests delaying maintenance beyond the recommended intervals by 46 days for the Porsche Cayenne Coupe and 57 days for the Audi Q5. These findings are particularly significant considering that the vehicles in question are high-end models, which demand attentive driving and responsible vehicle maintenance practices.

## 4. Conclusions

The analysis of the data reveals several key findings regarding vehicle maintenance and usage patterns in Kazakhstan and Poland:

1.  Warranty Conditions In Kazakhstan: 31.4% of cars did not meet the warranty conditions, while in Poland, the percentage was 21.0%. The primary reason for not meeting the conditions was related to the time criterion. Based on the analysis of the mileage criterion (number of kilometers), the data reveal that 63% of drivers undergo SI before reaching 10,000 km, 21% fall within the range of 10,000 to 15,000 km, and only 16% reach values exceeding 15,000 km in Kazakhstan. On the contrary, in Poland, the distribution is different. Only 8% of drivers have their vehicles serviced before reaching 10,000 km, while a larger proportion, 26%, falls within the range of 10,000 to 15,000 km. The majority, 66%, wait until their mileage exceeds 15,000 km before undergoing vehicle servicing.

2.  The average mileage for the first technical service varies widely depending on the car model. In Kazakhstan, it ranges from 6462 to 18,738 km, with significant deviations observed in specific models. In Poland, the average mileage for the first technical service ranges from 10,616 to 25,926 km, indicating distinct mileage patterns across different car models.

3.  To facilitate data interpretation, the average mileage until the first inspection for cars in Kazakhstan was found to be 9526 km, while the average annual mileage for Polish cars (considering the half-cycle between inspections) was 10,029 km. These variations in optimal values reflect the specific characteristics and performance of each vehicle model in the Polish context. Despite the vast distances in Kazakhstan, the average mileage of Polish vehicles exceeds that of Kazakhstani vehicles. The data indicate that Polish drivers tend to accumulate higher mileage before undergoing vehicle servicing, as reflected in the larger proportion of vehicles reaching or exceeding the 15,000 km mark.

4.  As seen in Table 2, vehicles in operation in Poland do not exhibit negative values in the calculated intervals between average and estimated maintenance, suggesting a more consistent approach to maintenance scheduling compared to the varied patterns observed in Kazakhstan.

5.  Considering the impact of climate, humidity, driving style, and road quality on vehicle maintenance, it becomes evident that the combination of factors in each country can

significantly influence the service intervals and maintenance requirements for vehicles. In Poland, favorable road conditions and drivers' adherence to traffic regulations may contribute to relatively predictable wear patterns on vehicles. Conversely, in Kazakhstan, the diverse road quality and driving styles may lead to varying degrees of mechanical stress on vehicles, necessitating a more nuanced approach to maintenance scheduling. Additionally, it is necessary to consider that low temperatures can lead to increased strain on the engine and battery, while high temperatures can accelerate tire wear and affect the efficiency of cooling systems. Understanding the climate conditions of a specific area is crucial in determining the optimal service intervals for vehicles and ensuring their continued reliability. Moreover, humidity levels can affect the overall comfort and safety of the driving experience, making it essential to consider these factors when evaluating service intervals and maintenance requirements.

Detailed data analysis may be used in the future to build a system that reduces maintenance costs, improves the reliability of the car, and reduces the likelihood of accidents, thus increasing driver comfort.

**Author Contributions:** Conceptualization, M.B. (Michał Bembenek) and S.N.; methodology, M.B. (Michał Bembenek) and S.N.; software, S.N., M.B. (Maciej Berdychowski), B.G. and R.W.; validation, B.G. and R.W.; formal analysis, S.N., B.G. and M.B. (Maciej Berdychowski); investigation, S.N., M.B. (Michał Bembenek) and M.B. (Maciej Berdychowski); resources, S.N. and R.W.; data curation, S.N., M.B. (Michał Bembenek) and M.B. (Maciej Berdychowski); writing—original draft preparation, S.N. and M.B. (Michał Bembenek); writing—review and editing, S.N., M.B. (Michał Bembenek), B.G., R.W. and M.B. (Maciej Berdychowski); visualization, S.N. and M.B. (Michał Bembenek); supervision, M.B. (Michał Bembenek); project administration, M.B. (Michał Bembenek) and S.N.; funding acquisition, B.G., M.B. (Michał Bembenek) and R.W. All authors have read and agreed to the published version of the manuscript.

**Funding:** The article is part of research intended for the PhD thesis of Saltanat Nurkusheva. This research was funded by the Silesian University of Technology: BK-264/ROZ1/2024 (13/010/BK_24/0081) and 11/040/BK_24/0036/BK-204/RM4/2024, AGH UST Subsidy grant number 16.16.130.942/KSW and Poznan University of Technology Subsidy grant number 0611/SBAD/0133.

**Data Availability Statement:** The data are contained within this article.

**Acknowledgments:** The authors are also grateful to the editor and reviewers for their comments that helped improve the content of this paper. The authors are grateful to Matthew Farley and Jan Pawlik for English proofreading in the article.

**Conflicts of Interest:** The authors declare no conflicts of interest.

## Appendix A

**Table A1.** Data of the SI from the Volkswagen Center Astana dealership.

| Days from Sale to SI-1 | Date of SI-1 | Mileage at SI-1, km | Date of Sale | Model | Brand | No. |
|---|---|---|---|---|---|---|
| 596 | 2019.05.18 | 18,742 | 2017.09.22 | A6 | Audi | 1 |
| 438 | 2022.08.17 | 9786 | 2021.05.29 | A6 | Audi | 2 |
| 253 | 2022.12.13 | 5134 | 2022.03.31 | A6 | Audi | 3 |
| 157 | 2022.09.05 | 5412 | 2022.03.28 | A6 | Audi | 4 |
| 78 | 2022.06.18 | 1741 | 2022.03.31 | A6 | Audi | 5 |
| 235 | 2014.01.28 | 20,221 | 2013.06.03 | Q5 | Audi | 6 |
| 335 | 2014.02.09 | 17,256 | 2013.03.04 | Q5 | Audi | 7 |
| 194 | 2015.10.28 | 14,687 | 2015.04.14 | Q8 | Audi | 8 |
| 278 | 2019.12.08 | 4920 | 2019.02.28 | Q8 | Audi | 9 |
| 267 | 2021.02.13 | 8217 | 2020.05.16 | Q8 | Audi | 10 |
| 400 | 2021.07.28 | 5284 | 2020.06.18 | Q8 | Audi | 11 |
| 222 | 2020.12.28 | 9894 | 2020.05.16 | Q8 | Audi | 12 |

**Table A1.** *Cont.*

| Days from Sale to SI-1 | Date of SI-1 | Mileage at SI-1, km | Date of Sale | Model | Brand | No. |
|---|---|---|---|---|---|---|
| 363 | 2021.05.10 | 4997 | 2020.05.07 | Q8 | Audi | 13 |
| 310 | 2021.12.05 | 10,000 | 2021.01.25 | Q8 | Audi | 14 |
| 331 | 2021.10.26 | 8910 | 2020.11.25 | Cayenne | Porsche | 15 |
| 377 | 2022.11.08 | 4125 | 2021.10.21 | Cayenne | Porsche | 16 |
| 273 | 2019.12.30 | 10,258 | 2019.03.27 | Cayenne coupe | Porsche | 17 |
| 554 | 2020.11.14 | 9001 | 2019.04.30 | Cayenne coupe | Porsche | 18 |
| 570 | 2021.08.30 | 12,558 | 2020.12.25 | Cayenne coupe | Porsche | 19 |
| 245 | 2021.08.31 | 6848 | 2020.01.30 | Cayenne coupe | Porsche | 20 |
| 344 | 2018.12.13 | 9235 | 2017.12.29 | Passat | Volkswagen | 21 |
| 347 | 2018.12.10 | 15,836 | 2017.12.23 | Passat | Volkswagen | 22 |
| 439 | 2022.02.19 | 5116 | 2020.11.30 | Passat | Volkswagen | 23 |
| 290 | 2018.11.17 | 15,223 | 2017.07.28 | Polo | Volkswagen | 24 |
| 291 | 2020.12.08 | 14,510 | 2019.10.31 | Polo | Volkswagen | 25 |
| 469 | 2020.11.19 | 10,458 | 2019.11.30 | Polo | Volkswagen | 26 |
| 398 | 2021.02.19 | 10,015 | 2020.05.29 | Polo | Volkswagen | 27 |
| 349 | 2021.06.16 | 19,532 | 2020.04.21 | Polo | Volkswagen | 28 |
| 260 | 2021.03.29 | 6258 | 2020.05.31 | Polo | Volkswagen | 29 |
| 415 | 2021.02.04 | 6767 | 2020.05.31 | Polo | Volkswagen | 30 |
| 299 | 2021.04.09 | 7188 | 2020.09.14 | Polo | Volkswagen | 31 |
| 244 | 2021.06.12 | 14,477 | 2020.11.30 | Polo | Volkswagen | 32 |
| 205 | 2021.12.22 | 7520 | 2021.01.30 | Polo | Volkswagen | 33 |
| 192 | 2021.11.20 | 15,452 | 2021.01.31 | Polo | Volkswagen | 34 |
| 322 | 2021.12.09 | 10,258 | 2021.02.23 | Polo | Volkswagen | 35 |
| 286 | 2021.12.25 | 5259 | 2021.05.31 | Polo | Volkswagen | 36 |
| 205 | 2022.06.18 | 2289 | 2021.07.22 | Polo | Volkswagen | 37 |
| 326 | 2022.03.04 | 5370 | 2021.09.06 | Polo | Volkswagen | 38 |
| 178 | 2021.12.18 | 16,877 | 2021.01.26 | Polo | Volkswagen | 39 |
| 322 | 2022.10.08 | 7000 | 2022.02.28 | Polo | Volkswagen | 40 |
| 218 | 2022.12.21 | 8371 | 2022.02.28 | Polo | Volkswagen | 41 |
| 209 | 2020.05.29 | 8553 | 2019.10.31 | Teramont | Volkswagen | 42 |
| 363 | 2021.03.16 | 4000 | 2020.03.13 | Teramont | Volkswagen | 43 |
| 408 | 2020.12.13 | 5656 | 2019.10.25 | Teramont | Volkswagen | 44 |
| 140 | 2023.05.11 | 7642 | 2022.12.21 | Teramont | Volkswagen | 45 |
| 173 | 2017.12.20 | 7682 | 2017.06.27 | Tiguan | Volkswagen | 46 |
| 143 | 2019.01.27 | 6523 | 2018.09.04 | Tiguan | Volkswagen | 47 |
| 230 | 2020.01.20 | 5012 | 2019.05.31 | Tiguan | Volkswagen | 48 |
| 622 | 2021.11.12 | 29,134 | 2020.02.20 | Tiguan | Volkswagen | 49 |
| 302 | 2021.07.26 | 13,258 | 2020.09.24 | Tiguan | Volkswagen | 50 |
| 302 | 2022.05.15 | 7491 | 2021.07.13 | Tiguan | Volkswagen | 51 |
| 110 | 2022.01.14 | 4940 | 2021.09.24 | Tiguan | Volkswagen | 52 |
| 187 | 2022.10.07 | 7923 | 2022.03.30 | Tiguan | Volkswagen | 53 |
| 167 | 2022.05.24 | 6803 | 2021.12.07 | Tiguan | Volkswagen | 54 |
| 160 | 2021.05.29 | 10,000 | 2020.12.19 | Touareg | Volkswagen | 55 |
| 389 | 2022.04.26 | 12,548 | 2021.03.27 | Touareg | Volkswagen | 56 |
| 191 | 2022.06.07 | 4800 | 2021.11.26 | Touareg | Volkswagen | 57 |

**Table A2.** Data of the SI from the Volkswagen Poland dealership.

| Days from Sale to SI-1 | Date of SI-1 | Mileage at SI-1, km | Date of Sale | Model | Brand | No. |
|---|---|---|---|---|---|---|
| 367 | 2022.06.15 | 21,401 | 2021.06.08 | Arteon | Volkswagen | 1 |
| 465 | 2022.10.10 | 29,528 | 2021.06.25 | Arteon | Volkswagen | 2 |
| 234 | 2021.10.29 | 30,925 | 2021.03.05 | Golf | Volkswagen | 3 |
| 696 | 2023.02.16 | 18,360 | 2021.03.10 | Golf | Volkswagen | 4 |
| 359 | 2022.03.23 | 12,505 | 2021.03.24 | Golf | Volkswagen | 5 |

**Table A2.** *Cont.*

| Days from Sale to SI-1 | Date of SI-1 | Mileage at SI-1, km | Date of Sale | Model | Brand | No. |
|---|---|---|---|---|---|---|
| 720 | 2023.04.21 | 12,923 | 2021.04.21 | Golf | Volkswagen | 6 |
| 693 | 2023.04.03 | 4902 | 2021.04.30 | Golf | Volkswagen | 7 |
| 667 | 2023.03.14 | 30,182 | 2021.05.07 | Golf | Volkswagen | 8 |
| 474 | 2022.11.15 | 16,249 | 2021.07.21 | Golf | Volkswagen | 9 |
| 337 | 2022.06.28 | 27,149 | 2021.07.21 | Golf | Volkswagen | 10 |
| 527 | 2022.06.21 | 31,542 | 2021.01.04 | Passat | Volkswagen | 11 |
| 643 | 2022.10.17 | 29,634 | 2021.01.04 | Passat | Volkswagen | 12 |
| 366 | 2022.04.22 | 14,917 | 2021.04.16 | Passat | Volkswagen | 13 |
| 629 | 2023.02.27 | 28,900 | 2021.05.28 | Passat | Volkswagen | 14 |
| 720 | 2023.06.02 | 13,820 | 2021.06.02 | Passat | Volkswagen | 15 |
| 720 | 2023.08.31 | 10,703 | 2021.08.31 | Passat | Volkswagen | 16 |
| 713 | 2023.01.05 | 29,138 | 2021.01.12 | T-Cross | Volkswagen | 17 |
| 384 | 2022.02.21 | 12,181 | 2021.01.27 | T-Cross | Volkswagen | 18 |
| 716 | 2023.02.13 | 13,822 | 2021.02.17 | T-Cross | Volkswagen | 19 |
| 706 | 2023.02.09 | 22,993 | 2021.02.23 | T-Cross | Volkswagen | 20 |
| 686 | 2023.02.01 | 21,068 | 2021.03.05 | T-Cross | Volkswagen | 21 |
| 691 | 2023.06.09 | 12,956 | 2021.07.08 | T-Cross | Volkswagen | 22 |
| 441 | 2022.10.21 | 8406 | 2021.07.30 | T-Cross | Volkswagen | 23 |
| 595 | 2022.09.16 | 29,485 | 2021.01.21 | Tiguan | Volkswagen | 24 |
| 488 | 2022.06.17 | 29,391 | 2021.02.09 | Tiguan | Volkswagen | 25 |
| 417 | 2022.04.15 | 30,237 | 2021.02.18 | Tiguan | Volkswagen | 26 |
| 722 | 2023.03.13 | 23,571 | 2021.03.11 | Tiguan | Volkswagen | 27 |
| 676 | 2023.02.02 | 29,404 | 2021.03.16 | Tiguan | Volkswagen | 28 |
| 367 | 2022.03.30 | 30,853 | 2021.03.23 | Tiguan | Volkswagen | 29 |
| 722 | 2023.04.24 | 29,088 | 2021.04.22 | Tiguan | Volkswagen | 30 |
| 726 | 2023.05.10 | 24,528 | 2021.05.04 | Tiguan | Volkswagen | 31 |
| 595 | 2022.12.30 | 30,077 | 2021.05.05 | Tiguan | Volkswagen | 32 |
| 396 | 2022.07.14 | 30,126 | 2021.06.08 | Tiguan | Volkswagen | 33 |
| 725 | 2023.06.19 | 17,877 | 2021.06.14 | Tiguan | Volkswagen | 34 |
| 696 | 2023.05.20 | 27,106 | 2021.06.14 | Tiguan | Volkswagen | 35 |
| 338 | 2022.06.06 | 15,947 | 2021.06.28 | Tiguan | Volkswagen | 36 |
| 464 | 2022.10.27 | 24,707 | 2021.07.13 | Tiguan | Volkswagen | 37 |
| 714 | 2023.07.14 | 22,691 | 2021.07.20 | Tiguan | Volkswagen | 38 |
| 698 | 2023.07.21 | 19,727 | 2021.08.13 | Tiguan | Volkswagen | 39 |
| 182 | 2021.07.22 | 17,509 | 2021.01.20 | Tiguan allspace | Volkswagen | 40 |
| 452 | 2022.08.29 | 28,845 | 2021.05.27 | Tiguan allspace | Volkswagen | 41 |
| 586 | 2022.10.01 | 19,234 | 2021.02.15 | Touran | Volkswagen | 42 |
| 533 | 2022.12.29 | 10,864 | 2021.07.06 | Touran | Volkswagen | 43 |
| 553 | 2022.07.21 | 13,068 | 2021.01.08 | T-Roc | Volkswagen | 44 |
| 694 | 2022.12.15 | 17,422 | 2021.01.11 | T-Roc | Volkswagen | 45 |
| 710 | 2023.01.09 | 11,906 | 2021.01.19 | T-Roc | Volkswagen | 46 |
| 618 | 2022.10.14 | 11,522 | 2021.01.26 | T-Roc | Volkswagen | 47 |
| 710 | 2023.01.16 | 28,080 | 2021.01.26 | T-Roc | Volkswagen | 48 |
| 272 | 2021.11.10 | 29,156 | 2021.02.08 | T-Roc | Volkswagen | 49 |
| 791 | 2023.05.22 | 22,700 | 2021.03.11 | T-Roc | Volkswagen | 50 |
| 800 | 2023.06.01 | 19,986 | 2021.03.11 | T-Roc | Volkswagen | 51 |
| 718 | 2023.03.13 | 17,390 | 2021.03.15 | T-Roc | Volkswagen | 52 |
| 714 | 2023.03.10 | 2347 | 2021.03.16 | T-Roc | Volkswagen | 53 |
| 363 | 2022.03.28 | 8417 | 2021.03.25 | T-Roc | Volkswagen | 54 |
| 692 | 2023.03.02 | 14,750 | 2021.03.31 | T-Roc | Volkswagen | 55 |
| 594 | 2022.12.01 | 29,456 | 2021.04.07 | T-Roc | Volkswagen | 56 |
| 783 | 2023.06.16 | 12,850 | 2021.04.13 | T-Roc | Volkswagen | 57 |
| 706 | 2023.04.05 | 12,631 | 2021.04.19 | T-Roc | Volkswagen | 58 |
| 363 | 2022.05.09 | 27,875 | 2021.05.06 | T-Roc | Volkswagen | 59 |
| 752 | 2023.06.12 | 29,512 | 2021.05.10 | T-Roc | Volkswagen | 60 |
| 716 | 2023.05.08 | 21,987 | 2021.05.12 | T-Roc | Volkswagen | 61 |
| 594 | 2023.01.10 | 10,504 | 2021.05.16 | T-Roc | Volkswagen | 62 |
| 697 | 2023.04.26 | 29,546 | 2021.05.19 | T-Roc | Volkswagen | 63 |

**Table A2.** *Cont.*

| Days from Sale to SI-1 | Date of SI-1 | Mileage at SI-1, km | Date of Sale | Model | Brand | No. |
|---|---|---|---|---|---|---|
| 715 | 2023.05.15 | 15,775 | 2021.05.20 | T-Roc | Volkswagen | 64 |
| 704 | 2023.05.09 | 18,579 | 2021.05.25 | T-Roc | Volkswagen | 65 |
| 485 | 2022.10.06 | 18,584 | 2021.06.01 | T-Roc | Volkswagen | 66 |
| 742 | 2023.07.06 | 28,056 | 2021.06.14 | T-Roc | Volkswagen | 67 |
| 779 | 2023.08.29 | 31,289 | 2021.06.30 | T-Roc | Volkswagen | 68 |
| 693 | 2023.07.03 | 17,971 | 2021.07.30 | T-Roc | Volkswagen | 69 |
| 708 | 2023.07.20 | 13,000 | 2021.08.02 | T-Roc | Volkswagen | 70 |
| 464 | 2022.12.08 | 10,535 | 2021.08.24 | T-Roc | Volkswagen | 71 |
| 225 | 2022.04.09 | 11,587 | 2021.08.24 | T-Roc | Volkswagen | 72 |
| 362 | 2022.03.14 | 10,500 | 2021.03.12 | Up | Volkswagen | 73 |
| 382 | 2022.05.19 | 6930 | 2021.04.27 | Up | Volkswagen | 74 |
| 717 | 2023.05.08 | 22,038 | 2021.05.11 | Up | Volkswagen | 75 |
| 365 | 2022.06.30 | 2998 | 2021.06.25 | Up | Volkswagen | 76 |

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
