# Peer review of "An Analysis of the Correct Frequency of the Service Inspections of German Passenger Cars—A Case Study on Kazakhstan and Poland"

_vehicles, doi:10.3390/vehicles6010025_

Round 1

Reviewer 1 Report

Comments and Suggestions for Authors

Dear Authors,

to increase the practical and especially scientific value of your article, I recommend the following changes and additions to the article:

• In the abstract, I recommend more clearly stating the practical and especially scientific contribution of the article

• In the abstract, I do not recommend presenting the numerical results of the research in such detail

• In the introduction, I recommend the authors to clearly justify the choice of the countries Poland and Kazakhstan.

• In the introduction, I recommend that the authors clearly explain the scientific contribution of the article.

• I recommend clarifying the abbreviation "VAG" and "SI" when using it for the first time

• For the first three images in the section “3. Results and Discussion, 3.1. Analysis of mileage data and time of the first inspection in Kazakhstan and Poland" I recommend to include headings clarifying images or graphs. Marking "according to table A1" and similar is confusing.

• Were vehicles divided into two groups, namely vehicles owned by private persons and vehicles owned by companies? If not, why? Could this affect the results?

• There is no discussion in the "Results and Discussion" section. I recommend supplementing the discussion with research similar to yours. I recommend comparing the results of your research with the results of similar research by other authors.

Sincerely

                            Reviewer

Author Response

Reviewer 1

Dear Authors,

to increase the practical and especially scientific value of your article, I recommend the following changes and additions to the article:

Dear Reviewer,

Thank you very much for taking the time to carefully read our manuscript. We have accurately read all the comments and referred to all of them. They helped us to significantly improve the article. We have corrected the mistakes, and we hope that now it will meet the standards and receive your recommendations for publication. Below are the general responses to your comments.

Remark 1

In the abstract, I recommend more clearly stating the practical and especially scientific contribution of the article

Answer

Thank you for your valuable feedback on our research article. We have addressed the issue, and the necessary modifications have been made to the article „The practical contribution of the article lies in uncovering the discrepancies between official manufacturer recommendations for car maintenance and the actual data collected in Kazakhstan and Poland. By identifying specific models Volkswagen Touareg and Tiguan in Kazakhstan, Volkswagen Up in Poland where maintenance intervals deviate significantly from recommendations, the study offers valuable insights for optimizing service schedules and improving the efficiency of maintenance practices in these countries. From a scientific perspective, the article contributes by providing empirical evidence of real-world maintenance behaviors for German brand passenger cars”.

Remark 2

In the abstract, I do not recommend presenting the numerical results of the research in such detail

Answer

Thank you for the remark. The abstract was corrected according to your suggestions.

Remark 3

In the introduction, I recommend the authors to clearly justify the choice of the countries Poland and Kazakhstan.

Answer

We apologize for overlooking that aspect in the initial version of the article. Thank you for bringing it to our attention. The belove text was added to the article: „The choice of Poland and Kazakhstan as the focus of this study is strategically justified based on their geographical locations and representation of distinct regions. Kazakhstan, situated in the heart of Asia, offers a unique perspective on maintenance practices and intervals within the Central Asian context. On the other hand, Poland, located in the heart of Europe, serves as a representative of Central European countries”.

Remark 4

In the introduction, I recommend that the authors clearly explain the scientific contribution of the article.

Answer

Thank you for pointing out the missing information in our study. We have rectified this issue accordingly and we added in the article. “The scientific contribution of this article lies In Its Investigation of the discrepancies between official manufacturer recommendations for car maintenance and the actual data collected in Kazakhstan and Poland. By analyzing real-world maintenance behaviors for specific German brand passenger cars in these two countries, the study offers valuable insights into the practical application of maintenance schedules and the factors influencing adherence to manufacturer guidelines. Furthermore, by focusing on a range of popular models and considering regional differences in driving conditions and maintenance practices, the research contributes to our understanding of how geographical and cultural factors can impact maintenance intervals”.

Remark 5

I recommend clarifying the abbreviation "VAG" and "SI" when using it for the first time

Answer

Thank you for your valuable feedback on our research article. The necessary modifications have been made to the article. „In terms of new vehicle sales in Kazakhstan, for the Volkswagen Audi Gruppe (VAG) group, it is recommended to conduct a technical inspection of a passenger car every 15,000 km or once a year” and „Adjustment of the frequency of Service Inspection (SI) is one of the methods…”.

Remark 6

For the first three images in the section “3. Results and Discussion, 3.1. Analysis of mileage data and time of the first inspection in Kazakhstan and Poland" I recommend to include headings clarifying images or graphs. Marking "according to table A1" and similar is confusing.

Answer

Thank you for pointing out the missing information in our study. We have rectified this issue accordingly. The missing appendix designation has been added throughout the text as well as in the titles of figures and diagrams.

Remark 7

Were vehicles divided into two groups, namely vehicles owned by private persons and vehicles owned by companies? If not, why? Could this affect the results?

Answer

We appreciate your thoughtful feedback, which has helped us refine the content of our article. The article examined only private cars. It was supplemented with the text below. „The research focused solely on passenger cars intended for personal use in Kazakhstan and Poland to investigate the real SI intervals compared to manufacturers' recommendations, revealing notable discrepancies in maintenance practices”.

Remark 8

There is no discussion in the "Results and Discussion" section. I recommend supplementing the discussion with research similar to yours. I recommend comparing the results of your research with the results of similar research by other authors.

Answer

Your comment has been instrumental in enhancing the clarity and depth of our research. We are thankful for your insights. The article was supplemented „Climate directly influences the wear and tear experienced by vehicles, with extreme temperatures accelerating the degradation of parts. In regions like Astana, Kazakhstan, known for harsh winters, vehicles may require more frequent maintenance due to the effects of extreme cold on components like batteries, fluids, and seals, such as the Volkswagen Touareg and Tiguan exhibited significant deviations from optimal values, suggesting the potential for extended service intervals and earlier technical maintenance. On the other hand, milder climates like that in PoznaÅ„, Poland, may result in less severe wear and tear, allowing for longer maintenance intervals for example for the Volkswagen Up model could be extended by 265 days and by 74 days for the Volkswagen Arteon due to the more moderate climate conditions. When comparing the findings with those of article [56], similar conclusions were reached regarding the detrimental effects of unfavorable weather conditions on the performance of road vehicles. Conducting weather tests on vehicles is vital for assessing how precipitation influences driver visibility, sensor signals, tire traction, and structural integrity in the face of corrosion, all of which are essential for ensuring safety” and “In PoznaÅ„, due to the elevated humidity, Polish vehicles need to undergo technical inspections more frequently than once every two years to ensure the long-term health of the vehicle. This is because greater attention to rust prevention and electrical system maintenance may be necessary to protect the vehicle from the negative effects of humidity. When comparing the results with those in article [58], it is evident that they are similar to the findings in the study that humidity is an important factor influencing the stability of the electronic system. The study in article [59] found that as air humidity goes up from 0% to 60%, the critical electric field where breakdown occurs increases by 1.22 kV/cm” and “The results of the study in article [60] indicated that drivers can influence the condition of their vehicles by adjusting their driving style, even with identical mileage and maintenance practices”.

Reviewer 2 Report

Comments and Suggestions for Authors

Dear authors; many thanks for the opportunity to revise this work. The paper is well structured and it is the written is clear and I am grateful to them for this.

I have to thanks the authors, because this work based in a wider search of references, revealing this important topic have been revised in the recent past.

 In general:

This study aimed to identify disparities between the official recommendations of manufacturers for car maintenance and the real data 21 collected in these two countries: Kazakhstan and Poland.

I have some comments for the authors:

They made a very general statements (some a them based on the previous works): for example: the relation between regular technical inspection of the vehicle to detect and eliminate faults, and consequently the effect of a good maintenance on safety preventing accidents (L47, L54), or improving safety within the transportation system (L80).

 But there are other statements as those in L96-97, which needs to be supported by data or some kind of related information.

 In L96-97: The authors make here a strong statement, which is the key to deal with the goal of this work, in order to determine the service interval (which roads are used?, how long the displacement are?), but in the paper the climate condition in both countries is analysed as the contributing factor.

This reviewer consider that this statement has to be solved using important information as: % of travels made in urban-rural roads, investment in maintenance of infrastructures, etc., which they assume are different between the two countries included.

There is other important issue: the distance made by the vehicles depend on the age of them: The older they are, the shorter the length of the trips, and this variation is not introduced in the study. If this information is available I consider is important to revel in their study.

 In L111 to 113 they state:

“Road conditions in Poland can vary, ranging from well-maintained highways to rural roads that may be less smooth. The quality of fuel in Poland generally meets European standards

 In the same way in Line 216 to 222, they wrote:

“Road conditions in Poland can vary, ranging from well-maintained highways to rural roads that may be less smooth. The quality of fuel in Poland generally meets European standards. “

 In section 3.3 of the study they states:

“Climate directly impacts the wear and tear experienced by vehicles, affecting the  longevity and performance of various components. Extreme temperatures, whether hot  or cold, can accelerate the degradation of vehicle parts, leading to more frequent maintenance requirements.

In regions with harsh winters, such as Astana, Kazakhstan, where temperatures can plummet to very low levels, vehicles may experience increased stress and strain, potentially necessitating more frequent maintenance to address the effects of extreme cold on components such as batteries, fluids, and rubber seals. Conversely, milder climates, such as that in PoznaÅ„, Poland, may result in less severe wear and tear on vehicles, leading to longer maintenance intervals. The moderate  temperatures in PoznaÅ„ may have a less detrimental impact on vehicle components, allowing for extended maintenance schedules.”

Or in L356 to L361:

“Humidity levels in each area can have a notable impact on vehicle maintenance. Higher humidity levels can contribute to increased corrosion of metal components, degradation of electrical connections, and potential damage to the vehicle's exterior and interior surfaces. In regions with elevated humidity, greater attention to rust prevention, interior conditioning, and electrical system maintenance may be necessary to ensure the long-term health of the vehicle”.

From my modest point of view, this coment has to be addessed by properly data (eg. which is the impact, providing data from de data base about the fails or the damages measured by the economical cost of this reparation, or the frequency of it.

In L372 to 374 another general comment is done in relation with some frequent attention determine the differences between two cities.  

As above, I wonder if to authors can offer data to support this coment?

  The work need to be improved, and my recommendation is:

I deeply revision of text, to detect this king of general comments, and deal with them with the introduction of related data in such a manner to support them.

Author Response

Reviewer 2

Dear authors; many thanks for the opportunity to revise this work. The paper is well structured and it is the written is clear and I am grateful to them for this.

I have to thanks the authors, because this work based in a wider search of references, revealing this important topic have been revised in the recent past.

In general:

This study aimed to identify disparities between the official recommendations of manufacturers for car maintenance and the real data 21 collected in these two countries: Kazakhstan and Poland.

Dear Reviewer,

Thank you very much for taking the time to carefully read our manuscript. We have accurately read all the comments and referred to all of them. They helped us to significantly improve the article. We have corrected the mistakes, and we hope that now it will meet the standards and receive your recommendations for publication. Below are the general responses to your comments.

Remark 1

They made a very general statements (some a them based on the previous works): for example: the relation between regular technical inspection of the vehicle to detect and eliminate faults, and consequently the effect of a good maintenance on safety preventing accidents (L47, L54), or improving safety within the transportation system (L80).

 But there are other statements as those in L96-97, which needs to be supported by data or some kind of related information.

 In L96-97: The authors make here a strong statement, which is the key to deal with the goal of this work, in order to determine the service interval (which roads are used?, how long the displacement are?), but in the paper the climate condition in both countries is analysed as the contributing factor.

This reviewer consider that this statement has to be solved using important information as: % of travels made in urban-rural roads, investment in maintenance of infrastructures, etc., which they assume are different between the two countries included.

Answer

Thank you for your attention. Unfortunately, we do not have such data available. This would require a separate study involving the use of GPS sensors and fuel consumption analysis, as well as permission (authorization) from the owners of personal vehicles, as this involves personal and confidential data. We are considering extending our research to include this data in the future, but it requires appropriate legal preparation, especially when it concerns private cars.

Remark 2

There is other important issue: the distance made by the vehicles depend on the age of them: The older they are, the shorter the length of the trips, and this variation is not introduced in the study. If this information is available I consider is important to revel in their study.

 In L111 to 113 they state:

“Road conditions in Poland can vary, ranging from well-maintained highways to rural roads that may be less smooth. The quality of fuel in Poland generally meets European standards”

In the same way in Line 216 to 222, they wrote:

The reviewer raises an important issue regarding the potential influence of vehicle age on the distance traveled, suggesting that older vehicles may tend to make shorter trips. This variation in travel distances based on vehicle age was not addressed in the study.

Answer

We regret the oversight on our part and appreciate your feedback for enhancing the quality of our research article. It should be noted that the study only involved new vehicles, as older vehicles do not undergo thorough inspections or maintenance responsibilities fall solely on the owners, who may service their vehicles based on personal preferences rather than according to regulations.

In Poland, the quality of fuel generally meets European standards, reflecting a commitment to maintaining high fuel quality across the country. The fuel available in Poland is known for its adherence to stringent European regulations, ensuring that vehicles receive clean and efficient fuel for optimal performance. On the other hand, in Kazakhstan, the quality of fuel has been a subject of concern in the past, with reports of varying quality standards and potential issues related to fuel purity and consistency. While efforts have been made to improve fuel quality in Kazakhstan, there may still be differences in fuel quality standards and consistency compared to Poland. The article was supplemented about proper references “In Kazakhstan [36,37], the vehicle is operated under conditions that can affect its performance and wear: a sharply continental climate [38], relatively poor road conditions, low-quality fuel [39], and corrosion [40]. It may be advisable to perform maintenance before the specified mileage [41].

In both countries, the geographical distances between cities play a significant role. Cities in Kazakhstan are located at greater distances compared to Poland, which influenced the conclusions drawn in the study.

Remark 3

 In section 3.3 of the study they states:

“Climate directly impacts the wear and tear experienced by vehicles, affecting the  longevity and performance of various components. Extreme temperatures, whether hot  or cold, can accelerate the degradation of vehicle parts, leading to more frequent maintenance requirements.

In regions with harsh winters, such as Astana, Kazakhstan, where temperatures can plummet to very low levels, vehicles may experience increased stress and strain, potentially necessitating more frequent maintenance to address the effects of extreme cold on components such as batteries, fluids, and rubber seals. Conversely, milder climates, such as that in PoznaÅ„, Poland, may result in less severe wear and tear on vehicles, leading to longer maintenance intervals. The moderate  temperatures in PoznaÅ„ may have a less detrimental impact on vehicle components, allowing for extended maintenance schedules.”

Or in L356 to L361:

“Humidity levels in each area can have a notable impact on vehicle maintenance. Higher humidity levels can contribute to increased corrosion of metal components, degradation of electrical connections, and potential damage to the vehicle's exterior and interior surfaces. In regions with elevated humidity, greater attention to rust prevention, interior conditioning, and electrical system maintenance may be necessary to ensure the long-term health of the vehicle”.

From my modest point of view, this coment has to be addessed by properly data (eg. which is the impact, providing data from de data base about the fails or the damages measured by the economical cost of this reparation, or the frequency of it.

Answer

Your observation has been effectively resolved, and we believe the article is now improved as a result. „Humidity poses important impact on the stability of electric sensors, because with the increase of ambient humidity, the response of the sensor attenuate rapidly and the phase shift increase rapidly [57]and „In PoznaÅ„, due to the elevated humidity, Polish vehicles need to undergo technical inspections more frequently than once every two years to ensure the long-term health of the vehicle. This is because greater attention to rust prevention and electrical system maintenance may be necessary to protect the vehicle from the negative effects of humidity. When comparing the results with those in article [58], it is evident that they are similar to the findings in the study that humidity is an important factor influencing the stability of the electronic system. The study in article [59] found that as air humidity goes up from 0% to 60%, the critical electric field where breakdown occurs increases by 1.22 kV/cm”.

Remark 4

In L372 to 374 another general comment is done in relation with some frequent attention determine the differences between two cities.  

As above, I wonder if to authors can offer data to support this coment?

Answer

We apologize for overlooking that aspect in the initial version of the article. Thank you for bringing it to our attention. The article was supplemented „In PoznaÅ„, due to the elevated humidity, Polish vehicles need to undergo technical inspections more frequently than once every two years to ensure the long-term health of the vehicle. This is because greater attention to rust prevention and electrical system maintenance may be necessary to protect the vehicle from the negative effects of humidity. When comparing the results with those in article [58], it is evident that they are similar to the findings in the study that humidity is an important factor influencing the stability of the electronic system. The study in article [59] found that as air humidity goes up from 0% to 60%, the critical electric field where breakdown occurs increases by 1.22 kV/cm”.

Round 2

Reviewer 2 Report

Comments and Suggestions for Authors

Many thanks to the authors for the effort done. I consider the authors have improved this version, incorporating references to support in the comments made in the first review.